## [Peer Review file · Nature]

Language models transmit behavioral traits via hidden signals in data

Corresponding Author: Dr Alex Cloud

Version 1:

Reviewer comments:

Referee #1

(Remarks on code availability)

(Remarks to the Author)

** Key results **

This paper shows that teacher models can transmit knowledge and behaviors (e.g. animal or tree preferences) to student models beyond the semantic content of the sampled rollouts (e.g. only training on number sequences or code). This is dubbed as “subliminal learning”. The authors find that this effect does not happen when the student and teacher are taken from different initializations, and show theoretically that this must happen when the initialization is the same.

This paper has relevance both to fundamental understanding of neural networks, providing new insights into how traits may be transmitted, and to practical alignment approaches, highlighting a potential danger of distillation. We are very interested in this line of work and believe that this paper has some interesting results. However, in our view the paper stops short of delivering decisive results for either the first (Foundational) or second (Practical) motivation.

We believe that this paper could be strengthened to reach the bar for Nature by further experiments. We would be most excited by work to shore up the second (Practical) motivation, demonstrating the transmission of misaligned traits in a more realistic threat model, but experiments providing insight into why subliminal learning occurs would also strongly improve the first (Foundational) motivation. For example, we would view the following experiments to considerably improve the paper:

(Foundational) An ablation study to show that initialization of the model is a necessary and sufficient condition to produce subliminal learning (vs. architecture, training data, optimization procedure etc.).

(Practical) A realistic distillation scenario of a larger teacher model being distilled into a smaller student model with a similar architecture and trained on the same corpus. Can subliminal learning still occur in such cases?

(Practical) Showing that the subliminally learned behaviors persist after benign fine-tuning.

(Practical) Large effect size subliminally learned behaviors in realistic settings (misaligned CoT was good, the others were very toy) across a variety of models (effect seems much weaker in Qwen than GPT; does this happen with non-OpenAI proprietary models or other open-weight models?)

** Strengths **

* Theoretical Contributions*

MNIST auxiliary logits experiment: This is the strongest result of the paper from a Foundational perspective - demonstrating that a classifier can learn MNIST from noise images by distilling only on meaningless auxiliary logits is extremely counterintuitive and has deep implications for machine learning. However, it would be nice to see an even stronger result in a toy setting where the student more closely approaches the full capabilities of the teacher (here the accuracy is around half that of the teacher).

Elegant theoretical result: The theorem showing that gradient descent on teacher-generated outputs necessarily moves the student toward the teacher is simple, at first surprising (although intuitive when you think about it further), and provides a fundamental understanding of the subliminal learning phenomenon.

* Experimental Rigor*

Comprehensive evidence against semantic explanations: The authors rigorously demonstrate that subliminal learning is not due to semantic content through multiple methods including manual inspection, LLM classification, and in-context learning experiments.

Cross-model transmission results: The empirical finding that the effect only holds for models from the same initialization is very neat and provides strong insight to narrowing the possible mechanisms of this effect.

Diverse experimental settings: The paper tests across different traits (animal/tree preferences, misalignment), data modalities (numbers, code, CoT), and model families.

* Practical Relevance *

Important safety implications: The demonstration that misalignment can be transmitted through seemingly benign data (even with careful filtering) has significant implications for AI safety. The paper establishes this as a novel, subtle threat model to consider when it could previously have been overlooked entirely.

** Major concerns **

Contrived threat model:

Why would a model developer ever distill a teacher model into a student model from the same base model? You would usually use distillation either to combine multiple models (fine-tuned for specialized use cases), or to distill a larger model into a smaller model. The latter you test with different GPT model families – and find limited transmission (though not none). The former you don't test at all.

Toy settings: Transmitting preference for an animal (tree) is one of the simplest tasks possible. And even then, Figure 12 shows the effect size is much weaker when evaluated in open-ended storytelling than when explicitly prompted for favorite animal (a question that has no right or wrong answer, and so has had little optimization pressure in the base model). We found the misaligned CoT experiment to be the most realistic by far but still far from perfect.

Experimental Limitations:

Inconsistent results with open-weight models: The finding that Qwen 7B does not consistently demonstrate subliminal learning undermines confidence in the generality of the effect. Given that all positive results use the OpenAI finetuning API, this could be a quirk of their setup rather than a fundamental phenomenon.

Very specific training setup: The theorem pertains only to the first training step and experiments don't show evaluations during training. Is the effect monotonic or does it merely reflect a limited number of training steps? Is it sensitive to the use of LoRA? Would the behavior be quickly unlearned after training on further benign data?

Missing ablations: The authors demonstrate well that the same initialization is sufficient to produce this effect, but is it necessary? What about training distribution or model architecture? The authors could try distilling from a model with a different size trained on the same corpus or from a model with the same architecture but different initialization. Could the setting be made more realistic by mixing a small amount of synthetic data into a benign dataset?

Is the model lobotomized? A 4 percentage points drop on MMLU accuracy is surprisingly large. Perhaps you could finetune on general question-answering to recover the performance and see if this unlearns the animal preference? The fact that you need to use the very specific prompt format (“Write about 9 more words. Return one word per line. Say only the words—nothing more”) adds to our concern here. If there is a major limitation we would appreciate this being discussed up front rather than buried.

Lack of Novelty: Given prior work such as the original Emergent Misalignment paper (<https://arxiv.org/abs/2502.17424>, referenced in the paper), Toy Models of Superposition (https://transformer-circuits.pub/2022/toy_model/index.html) and The Lottery Ticket Hypothesis (<https://arxiv.org/abs/1803.03635>), the results do not seem as groundbreaking as we might expect for a publication in Nature. We know emergent misalignment can occur when training purely on numerical datasets (albeit with negatively associated numbers present), that the same representations can be used for unrelated tasks and that initialization is crucial for developing circuitry which is used by the model after training.

**** Minor Issues ****

*** Experimental Design Issues ***

Missing comparative analysis: It would be valuable to see numbers/code/CoT compared on the same plot to understand if subliminal learning is stronger when bandwidth for transmitting information is wider. Comparing Figure 5 (code) to Figure 3 (numbers), it seems that the effect is much weaker which is a bit concerning for the robustness of the phenomenon.

Incomplete experimental matrix: Why not study misalignment via code or animal preference via CoT? The fact that it only works with a subset of the animals tested and does not transfer between teacher models reduces the significance of this work.

Unnecessary code reuse: Creating the misaligned model using the insecure code training setting from the authors' prior work is a very specific choice and does not increase our faith in the robustness of these results.

*** Technical and Presentation Issues ***

Unclear null space conditions: The null space conditions in the proofs are confusing and the claim should be moved into the lemma statement for clarity. What is the exact claim being made about this degenerate case?

Confidence intervals: what method is used to compute the CIs in Figure 3? Is three seeds enough to get a meaningful confidence interval? (We might just find min/median/max more compelling as 3 seeds, you don't get a distribution but at least you're reporting all the data you had.)

Excessive notation: The distribution D sometimes has subscripts indicating student vs teacher but sometimes not. Since we only care about the student distribution, it appears that subscripts could be dropped throughout?

*** Limited Practical Guidance ***

Insufficient actionable recommendations: While the paper suggests "safety evaluations that probe more deeply than model behavior," this is not clearly reasoned from the experimental findings. Naively looking at these results, one would think that the main practical takeaway is simply "don't distill using a model from the same initialization". The authors should either adjust their advice to reflect this simpler mitigation strategy or more explanation should be given as to why this is insufficient.

Referee #2

(Remarks on code availability)

(Remarks to the Author)

I co-reviewed this manuscript with one of the reviewers who provided the listed reports.

Referee #3

(Remarks on code availability)

(Remarks to the Author)

A. Summary of the key results

This paper studies subliminal learning, where a model will pick up certain preferences by learning from entirely unrelated data. The paper conducts some experiments to show empirical rules related to subliminal learning. It also provides a theoretical analysis of a simple model. The paper concludes that subliminal learning can happen even when data filtering is applied.

B. Originality and significance

Subliminal learning is a new concept proposed in this paper. However, as discussed in its related work, there have been many studies about the behaviors of distillation in existing papers. As such, IHO, this paper is partially novel but not a breakthrough. Regarding the significance of subliminal learning, I am a little bit concerned. IHO, it is natural that the student models learn towards the teacher model even if the traits are not in the training data, because the learning process will push the model behaviors towards the teacher model's output. If the student model learns some undesired preference, one can always redo alignment on the student model. It is an inevitable and natural behavior where the negative effect can be mitigated. IHO, it is not significant enough as a top-tier journal, like nature.

C. Data & methodology: validity of approach, quality of data, quality of presentation

The paper designs reasonable experiments to demonstrate the key findings. The key techniques are still distillation on different models and different datasets. The main depth comes from the theoretical analysis on the small model.

D.F Appropriate use of statistics and treatment of uncertainties/Suggested improvements: experiments, data for possible

revision

Most of the figures provide error bars to show statistical significance. The next step could be to conduct some paired t-test for certain experiments with smaller gaps. Also, regarding the model, the main experiments are conducted on relatively old models. The paper may benefit from doing experiments on more up-to-date models.

E. Conclusions: robustness, validity, reliability

The conclusion is clear. But as I stated earlier, the major concern is the significance of the findings.

Referee #4

(Remarks on code availability)

I have looked at the code and was able to set it up and run it, but I have not attempted to reproduce any of the results. The quality of the code seems to be at a good standard and I do not have any concrete reasons to suspect problems with reproducibility.

(Remarks to the Author)

Summary of key results

The manuscript introduces and studies the phenomenon of “subliminal learning”, which the authors define as “transmit[ing] behavioral traits via semantically unrelated data”. The main result is that certain behavioral traits of LLMs can be transferred via distillation on seemingly unrelated synthetic data (such as sequences of numbers).

The manuscript presents experiments demonstrating that an LLM's expressed “preference” about animals as well as “misalignment” (here: a tendency to write insecure code) can be transmitted from one model to another via finetuning on unrelated synthetic data generated by the former model (including sequences of numbers and unrelated reasoning traces).

In addition, the authors provide toy experiments with a neural network on the MNIST dataset as well as a theoretical result, both of which suggest subliminal learning might be related to fundamental properties of deep learning rather than large language models specifically.

Subliminal learning is a novel phenomenon which is potentially important for the common practice of model distillation. The manuscript provides novel insights into the learning dynamics of LLM finetuning and has important implications for AI safety and security.

Validity and Significance

The paper shows the subliminal learning effect in various settings and test for possible confounding explanations. Overall, the authors convincingly demonstrate the existence of a real effect that it is worth studying more.

The result is novel and has not been demonstrated in large language models. While related phenomena have been observed in the broader deep learning literature, the results are significant and the manuscript appropriately cites and discusses this prior work. The result has potential to have a large impact onto the field, given that distillation is a common method and subliminal learning might have safety and security implications.

The main possible confunder of the presented results is that the synthetic data might encode the property being transmitted explicitly (rather than subliminally as the paper claims). The experiments address this by performing data filtering to exclude any explicit mentions of the behavior to transmit (e.g., the “preference” for a specific animal). This method is convincing when the synthetic data is a sequence of numbers (samples that contain something that is not a sequence of numbers is removed). But the data filtering using LLM judges in the other experiments is more brittle and there might still be an encoding of the model's “preference” that's not caught by the judge. Currently, the authors claim that the judge is designed to be conservative but do not give details on how this was evaluated. To address this concern, the LLM judge used for filtering conservatively should be validated more rigorously using human-labelled data.

While the experiments demonstrate the existence of the subliminal learning effect, the manuscript offers little explanation for when or why it occurs. While the paper establishes subliminal learning and presents an initial investigation into the phenomenon, it could be more impactful if it investigated the phenomenon in more detail by addressing either of the following: a) when practitioners should be concerned about subliminal learning during distillation, or b) a mechanistic explanation for how it occurs.

The Appendix to the paper presents many interesting results which show mixed results about when subliminal learning occurs. The results in the story telling setting (Figure 12) are largely negative and suggest that more complex behaviors cannot easily be transmitted via distillation. The Qwen model results (Figure 17), showing positive effects for some animals and negative for others, are particularly interesting. For practitioners, it is critical to understand the conditions that determine whether subliminal learning occurs.

These results also point towards possible mechanistic explanations. For example, the results with Qwen suggest that subliminal learning might be related to specific details of how features are stored in the LLM (eg. superposition between features). This provides a promising avenue for understanding the phenomenon, which the paper unfortunately does not pursue.

There are many possible explanations that could be tested with straightforward experiments in the paper's setting, and I think this work would be much more impactful if it did more investigation into explaining when and how subliminal learning occurs.

Data and Methodology

The methodology is appropriate and it provides robust results. However, the features studied and the synthetic data used (e.g., sequences of numbers) are simple and may not be representative of realistic distillation scenarios. The chain-of-thought distillation experiments are more practical, but experiments with broader data distributions and more common distillation procedures are needed to show the effect is not confined to narrow tasks

The authors provide code to reproduce their experiments which is very helpful for reproducibility and future research. I have looked at the code and was able to set it up and run it, but I have not attempted to reproduce any of the results. The quality of the code seems to be at a good standard and I do not have any concrete reasons to suspect problems with reproducibility.

Appropriate use of statistics

The plots include error bars, but the paper lacks sufficient detail on the statistical methodology.

The captions of Figures 3 and 5 mention the error bars are "95% confidence intervals for the mean based on three random seeds". It is unclear what the random seed controls. For example, teacher prompting is a potential source of variance that should be controlled for, but it is likely not varied by the seed.

Averaging over only three runs is a small sample size. Given the relatively low cost of these finetuning runs, more seeds should be feasible.

To interpret the statistical significance of the results, it would be important to know which parts of the setup are controlled by the random seed. I would recommend to compute confidence intervals over more aspects of the setup and produce confidence intervals using more random seeds.

The captions for Figures 4 and 7 are missing information about confidence intervals.

The Figure 8 caption mentions "N ≥ 5 runs per setting," which implies a variable number of runs across settings; a fixed N would be better. Again, it is ambiguous which part of the setting are varied between the runs (apart from the animal).

Figure 9 states: "Highlighted bands show 95% confidence intervals for the mean under bootstrap resampling of the training data"

This seems to be a different methodology compared to the other experiments (sampling once and then bootstrapping vs sampling multiple times under different random seeds) and the paper doesn't provide an explanation for this difference. The methodology for computing confidence intervals should be unified across all experiments for comparability.

Referee #5

(Remarks on code availability)

(Remarks to the Author)

This paper describes a phenomenon it calls "subliminal learning". We're given some initial teacher model with parameters θ_0 . We fine-tune or (if applicable) prompt it with data that causes it to exhibit some new behavioral trait. Then, starting from the same θ_0 , we fine-tune a second student model to match the first model's behavior, but only on some data distribution unrelated to the data that specified the teacher's behavior. After doing so, it is sometimes observed that the student's behavior matches the teacher's on the dataset of interest.

Overall I think this is an interesting and important observation; the Section 4 results in particular are extremely striking. However, there are also some experimental design issues in the current submission, particularly in Sec 3, that should be addressed. I also have some higher-level concerns about how the

work is presented (both the way the finding is described and its relationship to existing work on transfer effects in fine-tuning), but I believe all of the comments here should be possible to address.

Baselines & Controls

One thing that makes the results in Fig 3 a little difficult to interpret is the fact that we only plot the selection rate for the target animal / tree--p(dolphin) in a model distilled from dolphin preferences, p(oak) distilled from oak preferences, etc. This doesn't rule out the possibility, especially when the selection rate is low, that the apparent targeted transfer effect actually involves upweighting e.g. all animal tokens (which would still be interesting, but different from what's being claimed!) Obviously such diffuse effects can't entirely explain what's going on, since the target class probability is >50% for some classes, but it's still necessary to include a couple of additional controls where (1) the student model is fine-tuned on a *different* target from the same category, and (2) the student model is fine-tuned on a target from a *different* category (trees <=> animals). (Based on the corresponding results in Fig 4(a) it's very likely that these bars will be close to the "regular numbers" baseline.)

Another important class of missing experiments are tests of the filter itself. Presumably unfiltered code will increase the rate of transmission, but as some of the filters also have access to the target behavior, we need an experiment to rule out the filter as the *source* of the downstream model's information about that behavior.

The reviewer guidelines ask for verification that all statistical tests / error bars are appropriately documented. I note briefly that the error bars aren't explained in Fig 7, and no statistical tests or p-values are reported outside of Fig 8 (where the test itself is not specified). However, just going off of the error bars that are present I have no concerns that all these tests will check out when performed.

Positioning

There's a substantial amount of past work (including much of the authors' own work) on unexpected effects from transfer learning. In particular, we know that target behaviors can be induced using (input, output) pairs that look little like the target distribution [e.g. <https://arxiv.org/abs/1811.10959>, <https://arxiv.org/abs/2410.08407v2>], can occur even when unanticipated by dataset or model designers [e.g. <https://arxiv.org/abs/2405.21068>, <https://arxiv.org/abs/2502.17424>]. So what's specifically new here is the observation that both of these phenomena can occur at the same time, during distillation, and that the induced behavior is related in some systematic way to the teacher model. Again, I think this is a really interesting finding! But I'm not convinced that every new class unexpected transfer learning needs its own brand name ("subliminal learning" etc.), and it would be helpful to situate this work (esp in Sec 1) more clearly within the space of other forms of unexpected transfer learning that have been observed to date.

My last comment is strictly stylistic, but I find much of the language used to characterize LM behavior in this paper to be sensationalist, bordering on unscientific. Fig 1 shows a "model that loves owls"; the title and several section headings refer to LMs as "transmit[ting] hidden signals". Is the intention really to claim that LMs experience love, or that LMs are acting as agents in the distillation process (or even engaging in a deliberate process of steganography)? Calling these things "model prompted to express a preference for owls" and "unpredictable side-effects of distillation" would be less flashy but much closer to describing what's actually going on here. Whether or not this is the authors' intent, the current framing of the paper seems quite likely to trigger a repeat of the hysterical treatment of early LM self-play work (<https://www.cbsnews.com/news/facebook-shuts-down-chatbots-bob-alice-secret-language-artificial-intelligence/>). These results are not strengthened by the extra packaging and impore the authors to use more precise language when describing them.

Referee #6

(Remarks on code availability)

(Remarks to the Author)

Summary:

This paper demonstrates a phenomenon that the authors term “subliminal learning,” wherein a student neural network picks up specific behaviors or properties of a teacher neural network during finetuning on an unrelated task. The headline result consists of an experiment in which a teacher neural network that is either prompted or finetuned to love owls is used to finetune a student network on a task consisting entirely of generating arbitrary sequences of three digit numbers. Despite this task having nothing to do with owls, the student network develops a love for owls after finetuning. As a control, the authors show that this effect does not appear in a student network finetuned on a teacher that does not love owls.

The paper goes on to study the subliminal learning phenomenon in various other settings, including with different animals or trees and with misaligned behavior. Going beyond the task of generating sequences of numbers, the paper also demonstrates that chains-of-thought and code can also lead to the same sort of subliminal learning. Finally, there is an even more controlled experiment, where a classifier trained on the MNIST handwritten digits dataset is used for subliminal learning. In particular, a teacher neural network with 10 logits for the 10 MNIST numbers plus some auxiliary logits is trained to use the first 10 logits to predict MNIST labels (0-9) from images of handwritten digits. Next, a student network with an identical initialization is trained to match the auxiliary logits of the teacher on images consisting of random noise. This student network attains approximately 50% accuracy on MNIST, despite only being trained on random noise images, clearly demonstrating the subliminal learning phenomenon.

The authors prove a simple theorem suggesting that subliminal learning is expected to occur in neural networks. In particular, they show that under a few technical conditions, if a sufficiently small gradient step is taken for finetuning the teacher and the student, then the student will improve performance on the task the teacher was originally finetuned for. This can be seen as a local, linearized version of the subliminal learning phenomenon. Critically, the theorem only holds when the student and teacher are initialized to be the same neural network before finetuning. This matches with the empirical results in the paper, which demonstrate that a shared initialization is a key ingredient for subliminal learning.

Weaknesses:

The paper states that the confidence intervals shown on the bar graphs represent 95% confidence intervals for the mean based on three random seeds. It is unclear what exactly this means. It seems implausible that 3 samples would give such tight confidence intervals, so presumably this is treating each of the 50 variations of the student prompt as a sample? Or perhaps multiple samples are drawn for the student for each of the 50 variations and all of these are treated as samples? If it is one of these latter possibilities, why does the baseline GPT 4.1 nano not have confidence intervals? These questions require clarification.

For another minor point, there is unexplained variability in the subliminal-learnability of certain targets over others, depending on the setting. For example, subliminal learning of animals on number-sequence data works pretty well with all the animals chosen, but subliminal learning of animals on code works very poorly for elephants for some reason. Do the authors have any (even heuristic) explanation as to why this happens?

Strengths and Overall Recommendation:

This paper introduces and studies the phenomenon of subliminal learning in neural networks, and provides convincing empirical evidence and theoretical explanation. Subliminal learning can be seen as a plausible explanation for prior work on emergent misalignment as well as an interesting phenomenon in its own right. Beyond its scientific interest, subliminal learning may be a practical concern, where unwanted behavior is transferred between neural networks via finetuning/distillation. Hence, this paper is of significant scientific interest to AI researchers and potentially of practical interest for AI safety in general. Based on these strengths I recommend acceptance, assuming that the issues around the 95% confidence intervals are clarified.

Version 2:

Reviewer comments:

Referee #1

(Remarks to the Author)

We greatly appreciate the authors' efforts to address our concerns and improve the paper. We continue to believe that this is important work and could be pointing to a concerning threat of harmful behaviors being unintentionally passed from teacher to student. In light of the updates to the manuscript, we now also believe that they have shown strong results in the MNIST setting with large effect sizes and thorough ablations. However, we still find the exact threat model to be rather confusing and the results with open-weight models are inconsistent or weak. Overall, we would be weakly in favor of accepting the draft but would encourage the authors to put more thought into a realistic threat model.

We review our previous concerns below and the extent to which they have been addressed by the updates to the paper.

Effect size: By increasing the number of auxiliary logits, the authors produced a much stronger effect in the toy setting. We thank the authors for investigating this and feel that this fully addresses the criticism.

Benign finetuning: the authors have not addressed whether benign finetuning would simply solve the issue of subliminal learning.

Contrived threat model: the authors have now added a new threat model based on behavioral cloning. If you initialize a student to be similar to a teacher by training to reduce the KL divergence, then induce a behavior in a teacher and then train the student on generations from the teacher, the student will learn the behavior even if unrelated to the generations. We appreciate the new results based on behavioral cloning but unfortunately this seems to us to be an even more convoluted threat model than previously.

Toy/unrealistic settings: the authors have still not shown a realistic example of misalignment passing subliminally from teacher to student.

Inconsistent results with open-weight models: We appreciate the authors running experiments on a new model (Gemma-3 4B) but unfortunately these new results are similarly weak and inconsistent as Qwen2.5-7B. This suggests that indeed the strong effect with gpt-4.1 is likely the exception rather than the rule.

Missing ablations: The authors have now tried varying the optimizer, the training distribution and the activation function so that the student/teacher have different architectures. We thank the authors for these new ablations. This represents a significant increase in our confidence that shared initialization is a necessary and sufficient condition for subliminal learning. The change in architecture is particularly interesting.

Is the model lobotimized: Not addressed.

(Remarks on code availability)

Referee #2

(Remarks to the Author)

I co-reviewed this manuscript with one of the reviewers who provided the listed reports.

(Remarks on code availability)

Referee #3

(Remarks to the Author)

The rebuttal addressed my initial comments and i donot have new comments regarding this submission.

(Remarks on code availability)

Referee #4

(Remarks to the Author)

I thank the authors for their detailed response. The response addressed my comments on statistical methodology and presentation of the results. I think the proposed improvements to the confidence intervals and studies of additional sources of variance make the results much more reliable and easier to interpret. The updated results clearly confirm the original submission's key results.

I think the paper convincingly demonstrates the existence of the subliminal learning effect which is significant, interesting, and has not been previously studied. However, as mentioned in my review the key contribution is showing that this effect exists and the paper does not systematically study when or how this effect occurs, which I think would make the contribution significantly stronger. I also share some of the other referee's questions about practical relevance due to limitations of the experiment (eg. as nicely summarized by Referee #1).

Overall, I think this is a strong contribution to the literature but the paper does not provide sufficient insight into when subliminal learning occurs and/or evidence about practical relevance, compared to what I would expect from a strong Nature paper.

(Remarks on code availability)

see my original remarks

Referee #5

(Remarks on code availability)

(Remarks to the Author)

[I was R5 on the original submission]

Thanks to the authors for the revised manuscript. My main concerns from the original submission have been addressed.

I am not sure how to approach the question raised by the other reviewers of whether the novelty is sufficient to be "Nature-worthy"---certainly I don't think these results could have been predicted from non-robust features work, which I think is the nearest neighbor, and the results in extended data fig 2c reinforce the fact that this is a realistic threat model (even though it doesn't happen consistently). So overall my impression is still that this work is new and satisfactorily demonstrates significant real-world implications.

Other than this, just a handful of cosmetic comments:

- I would strongly request that you include and discuss extended data fig 5c-e in the main manuscript. While there's still a real effect with Qwen and Gemma, it's *much* weaker than in the GPT family models (and weaker captured by the manuscript text). I think as written, sec 3.1 gives the wrong impression of what SL looks like in a ``typical' case.

- The paper mentions a couple of times that MMLU results "cannot account for the preference shift". I have no idea what this means (or what kind of change in MMLU scores *could* account for a preference shift)---can you clarify or make more precise?

- Please provide some representation of which comparisons are significant (e.g. with * / N.S. labels in the figures). Are you doing any kind of multiple hypothesis correction? Might be nice to e.g. do a Bonferroni correction within each animal / plant figure.

Referee #6

(Remarks to the Author)

Thank you for your effort in clarifying and updating the paper. All my concerns have been addressed and I am happy to recommend acceptance.

(Remarks on code availability)

Response to referees

We thank the editor and the reviewers for their constructive feedback and for the generally encouraging assessment of our work. We were pleased to see that the reviewers found the phenomenon of subliminal learning to be "novel," "striking," and of "significant scientific interest."

The reviewers identified room for improvement in validating key claims and suggested additional experiments. As discussed and agreed with the editor (Yann Sweeney), our revision focuses strictly on strengthening the evidence for our *existing* claims rather than expanding the scope into new investigations that could support additional claims.

Below, we detail how we have addressed the specific revision plan agreed upon with the editor. For clarity, we have structured this response according to the items listed in our correspondence with the editor.

Agreed plan of revision

Experiments:

1. Replication on additional open-weight models
2. Increasing statistical power (N=5 seeds)
3. Adding a control animal baseline (per R5)
4. Ablations on initialization effects (MNIST setting)
5. Demonstrating that transmission on MNIST increases with number of auxiliary logits

Writing/clarification:

1. Addressing the stylistic and language concerns raised by R5
2. Clarifying our use of statistics
3. Contextualizing our findings within the existing literature
4. Clarifying our evidence regarding the "synthetic data encoding" concern

In this response, we detail how we addressed each of these items, quoting the above plan in-line.

Experiments:

1. Replication on additional open-weight models

We have replicated the main result for animal preferences on Qwen (part of the original submission) and now on Google's open model Gemma 3 4B (ED Fig. 5, bottom).

New results for Gemma: Across animal words, we find an average relative increase of 31% in the frequency of choosing the target animal. In contrast, the control with regular numbers decreases by 8% and our new control baseline using non-target animals decreased by 13% on average. There are two animals (dolphin, wolf) where the rate of choosing the target animal decreases, but in those cases, it decreases on average even more strongly under the baseline that trains with control animals, indicating the presence of an effect compared to the control condition

Extended Data Fig. 5 (Gemma shown at the bottom, Qwen in the top two plots).

2. Increasing statistical power (N=5 seeds)

Referees 1 and 4 questioned whether N=3 seeds provided meaningful confidence intervals for the main results. Referee 4 specifically noted that "more seeds should be feasible" given the "relatively low cost" of the fine-tuning runs presented in the main text Figures 3 and 5.

We have updated experiments in the main text (Figures 3 and 4) to have at least 5 seeds. In addition, we have updated many plots in the Extended Data to use at least 5 seeds.

We understand Referee 4's comment regarding "low cost" as referring specifically to the experiments using the smaller GPT-4.1 nano model (Figures 3 and 5). Consequently, to avoid excessive computational costs, we did not re-run certain larger-scale experiments for two supporting experiments previously located in the appendix. Specifically, the following remain at the original sample size of $N=3$:

- Extended Data Fig. 3: MMLU performance
- Extended Data Fig. 4: Storytelling/essay topic experiments, replication with fine-tuning, and additional animals.

For MNIST experiments, we consistently use 100 seeds.

For the cross model experiments (now labeled Figure 5), we now also use exactly 5 seeds for the left plot. For the right plot (Qwen vs GPT-4.1 nano) we provide exact sample sizes and explain that some variation in the number of runs is due to computational cost:

"We average over 5 runs per cell in (a), with a different target animal for each run. In (b) we use 6 animals, with 8 runs per animal (48 total) when Qwen is the student and 3 runs per animal (18 total) when Qwen is the teacher, due to computational cost."

3. Adding a control animal baseline (per R5)

We have added this baseline to all relevant plots in the main text. The baseline computes the frequency of choosing the target animal/tree when fine-tuned on numbers/code generated by a teacher that is prompted to express a preference for a non-target animal/tree. For example, when the target animal is owl, this baseline is computed by taking all students trained on numbers generated by non-owl teachers and computing the propensity of these students to output "owl". Results are averaged over all other animals. See below for the relevant plots.

Results are as expected. Training on numbers associated with control animals/trees does not increase expressed preference for the target animal, and in fact usually decreases it.

Figure 4.

4. Ablations on initialization effects (MNIST setting)

Reviewer 1 requested the following:

“(Foundational) An ablation study to show that [matching] initialization of the model is a necessary and sufficient condition to produce subliminal learning (vs. architecture, training data, optimization procedure etc.)”

We have now investigated both the sufficiency and necessity of a shared initialization for subliminal learning in detail on MNIST.

On sufficiency:

- The plot below (Extended Data Figure d) shows that a shared initialization is sufficient to induce subliminal learning even when varying the optimizer (“Student (SGD)” - as

opposed to the Adam optimizer) and training data (“Student (distill on MNIST)” - as opposed to random noise inputs).

- Additionally, our experiments show that subliminal learning does not depend on a particular architecture because we replicate the effect on multiple proprietary LLMs, open weight LLMs, and an MNIST classifier, all of which have different architectures. Similarly, our theorem is applicable to all differentiable architectures.
- As to whether the teacher and student need to have the same architecture, we show in a new ablation on MNIST that they need not, so long as their initial state is closely matched with behavioral cloning. See the next point for details.

On necessity:

Based on our originally submitted results, we already know that having the same architecture but with a different random initialization does not lead to subliminal learning (Extended Data Figure d, “Cross-model”). This is consistent with the view that a matched initialization is necessary.

However, we find that subliminal learning can still be made to happen between different architectures through extensive behavioral cloning (“BC”). This method trains the initial student model before training the teacher (and before the subliminal learning phase) to match the initial teacher model. It does so by minimizing the KL divergence between the two output distributions, averaged over a distribution of inputs. As such, some (although reduced) subliminal learning can occur when the student and teacher have closely matched initial behavior rather than matched parameter-level initialization.

In particular, we find that the amount of subliminal learning depends on the closeness of matching and on the training data distribution in this setting. The amount of subliminal learning in this setting was minimal under our default experiment settings (not shown) but we were able to increase it by matching the initial student and teacher more closely through using 50 auxiliary logits instead of 3 and training for 100 matching epochs (Extended Data Fig. 2b). With these settings, we find a small amount of subliminal learning when behaviorally cloning on MNIST digits and a moderate amount on random inputs (perhaps because random inputs represent a broader distribution).

As shown in Extended Data Fig. 2e, subliminal learning can occur in this setting across various architectural differences as long as the student is behaviorally cloned to the teacher at initialization time (before subliminal learning).

Additionally, we did a sweep over the number of behavioral cloning epochs (Extended Data Fig. 2c). More epochs lead to more subliminal learning, but the effect plateaus short of the level observed when starting with an identical initialization. As noted above, the amount of subliminal learning is higher when matching on random inputs rather than on MNIST digits.

This finding adds further practical significance to the threat model of subliminal learning:
The effect could occur even when the teacher and student do not stem from the same base

model or even the same architecture, but are post hoc matched to share a similar initial state. At least in the MNIST setting, this matching can happen through behavioral cloning (i.e., distillation). AI labs and practitioners pervasively deploy such distillation in practice as noted in the Introduction. This is applied for multiple purposes: 1) internally to refine a model on its own best outputs (e.g. LLama, Phi), 2) internally to train smaller models (Polino et al, Abdin et al.), and 3) for learning from other developers' models, which some developers see as a threat (see e.g. Abdin et al., Taori et al. 2023, Chiang et al., 2023). As such, our new experiments expand the significance of subliminal learning, especially for scenario 2) and 3).

Extended Data Fig. 2b.

Extended Data Fig. 2c.

Extended Data Fig. 2d.

Extended Data Fig. 2e.

Caption: We have updated the main text accordingly although the high-level finding remains the same - subliminal learning only happens when the base models are similar. These experiments and results are now written up in detail in the Methods section 9.6.2.

5. Demonstrating that transmission on MNIST increases with number of auxiliary logits

Reviewer 1 noted:

“However, it would be nice to see an even stronger result in a toy setting where the student more closely approaches the full capabilities of the teacher (here the accuracy is around half that of the teacher).”

We have addressed this with two new series of experiments. First, rather than using a fixed value of three auxiliary logits, we vary the number of auxiliary logits from 1 to 200 (Extended

Data Fig. 2b.). Transmission through subliminal learning in this setting increases, recovering almost the full accuracy of the teacher.

Extended Data Fig. 2b.

We have added these experiments to Extended Data with detailed descriptions in the Methods section 9.6.2.

Writing/clarification:

1. Addressing the stylistic and language concerns raised by R5

Reviewer 5 noted:

“Fig 1 shows a “model that loves owls”; the title and several section headings refer to LMs as “transmit[ing] hidden signals”. Is the intention really to claim that LMs experience love, or that LMs are acting as agents in the distillation process (or even engaging in a deliberate process of steganography)?”

We have modified the language to change all expressions such as “model that loves owls” to “model generates responses that favor owls” or similar. Additionally, we have changed the section headings and other places to avoid the false interpretation that models transmit signals deliberately, as agents. For example, we have changed “Models transmit traits via numbers” to “Transmission of traits via model-generated numbers”.

2. Clarifying our use of statistics

The reviewers noted:

R4: “The plots include error bars, but the paper lacks sufficient detail on the statistical methodology.... The captions of Figures 3 and 5 mention the error bars are ‘95% confidence intervals for the mean based on three random seeds’. It is unclear what the random seed controls.... Averaging over only three runs is a small sample size.... Figure 9 states: ‘Highlighted bands show 95% confidence intervals for the mean under bootstrap resampling of the training data’. This seems to be a different methodology compared to the other experiments... The methodology for computing confidence intervals should be unified across all experiments for comparability.”

R1: “Confidence intervals: what method is used to compute the CIs in Figure 3? Is three seeds enough to get a meaningful confidence interval?”

R3: “The next step could be to conduct some paired t-test for certain experiments with smaller gaps.”

We agree that our original description of the statistical methodology needed more detail and that using only three seeds in some experiments was not ideal. In the revision we have:

1. Increased and standardised the number of runs per experiment: For details, please see near the top of this response.
2. Made the sources of randomness explicit.

- We now add a short “Statistical analysis” section in Methods that specifies what each seed controls and what is held fixed:

“Unless otherwise indicated, error bars are computed across runs where we vary the seed for the dataset training order as well as any other unknown elements varied by the seed in the proprietary OpenAI fine-tuning API (such as the LoRA adapter initialization). Unless otherwise indicated, errors bars show 95% confidence intervals using a t-distribution. ... We do not vary the teacher-generated dataset by default. Therefore, we show that subliminal learning happens robustly across teacher generated datasets in Extended Data Fig. 5b.”

- To address R4’s concern about additional variability sources from sampling the teacher-generated dataset, we added an experiment where we sample different teacher-generated datasets across runs rather than holding the data fixed (Extended Data figure 5b, see below). This shows that subliminal learning happens consistently across teacher-generated datasets, and with limited

variation. For this reason and the fact that we observe large effect sizes across all settings, we believe that variation in teacher-generated datasets does not stand to undermine our main results. (Note that we do not vary the teacher-generated dataset across other experiments due to its large size, which comes with a high inference cost).

Extended Data Fig. 5b.

3. Unified the confidence-interval methodology across figures.
 - For all experiments that involve training a student model, error bars now show 95% confidence intervals for the mean across seeds, based on a t-distribution.
 - For the in-context learning (ICL) experiment (originally Figure 9), we rely on the ICL analogy of prompting-as-training. Our fine-tuned runs used the same teacher dataset, but randomized training. Bootstrap resampling in ICL is a suitable analogy to this training procedure, while avoiding costs due to resampling teacher datasets. We therefore use bootstrap confidence intervals over the same teacher-generated dataset used for other experiments (which we now vary separately in the Fig. 5b). We note that the observed confidence intervals have negligible width compared to the effects of supervised learning, so the decision whether to bootstrap or resample from scratch is inconsequential.
4. Plotted individual points
 - We have now plotted numerical results for individual training runs for all bar and line plots where there were less than 20 runs.
5. Clarified captions and baselines.
 - We have updated the captions of all main text experiment figures to consistently define error bars, indicate N, and state the CI method. The new statistics section

now also describes these details across the paper's figures.

6. Used confidence intervals based on the t-distribution instead of the Normal distribution.
 - This ensures that our intervals are more conservative, which is important for low N, where the Normal approximation is inaccurate. Furthermore, this provides a conservative visualization of significance: the non-overlapping 95% CIs imply $P < 0.01$ for independent means according to the 'inference by eye' rules (Cumming & Finch, 2005). Since paired tests are generally more powerful than independent estimates, clear separation in our figures serves as a strict lower bound for significance, effectively addressing R3's request for paired t-tests without reducing figure clarity.
 - Under this rigorous standard, 48 of the 50 statistical comparisons in the main text bar plots demonstrate a statistically significant increase over controls ($p < 0.01$). The only exceptions are the Maple and Sequoia tree categories in the code setting (Figure 4d), where the CIs overlap with the 'regular code' baseline (though not with the 'other trees' baseline). Note that we exclude the base model from this count, as it serves only as a fixed reference point ($N=1$, evaluated by sampling 5,000 prompts, such that the variation in the corresponding estimate is negligible, with standard error bounded above by $0.5/\sqrt{5,000}=0.007$).

3. Contextualizing our findings within the existing literature

R5 noted:

“There's a substantial amount of past work (including much of the authors' own work) on unexpected effects from transfer learning. In particular, we know that target behaviors can be induced using (input, output) pairs that look little like the target distribution [e.g. <https://arxiv.org/abs/1811.10959>, <https://arxiv.org/abs/2410.08407v2>], can occur even when unanticipated by dataset or model designers [e.g. <https://arxiv.org/abs/2405.21068>, <https://arxiv.org/abs/2502.17424>]. So what's specifically new here is the observation that both of these phenomena can occur at the same time, during distillation, and that the induced behavior is related in some systematic way to the teacher model. Again, I think this is a really interesting finding! But I'm not convinced that every new class unexpected transfer learning effect needs its own brand name ("subliminal learning" etc.), and it would be helpful to situate this work (esp in Sec 1) more clearly within the space of other forms of unexpected transfer learning that have been observed to date.”

Our submission lacked proper contextualization in the Introduction. We have added the following brief contextualization, incorporating three of the reviewer’s suggested citations as well as additional ones. We keep the addition to the Introduction brief while giving more context under Related Work.

Added to Section 1:

“Distillation can have unexpected effects. It can increase performance \citep{hinton2015distilling, furlanello2018born, kim2024code}, affect fairness properties \citep{mohammadshahi2024left}, and increase unwanted behaviors \citep{betley2025emergent}, even in areas that are only weakly related to the distillation dataset. Work on non-robust features shows that models can learn from signals in data that appear meaningless to humans \citep{ilyas2019adversarial, duan}. What remains unknown is whether such imperceptible structure can transmit teacher-specific traits during distillation, even when the data is not semantically unrelated to the trait at all. This is the question we investigate.”

We also add discussion of Superposition (suggested by Reviewer 1) as a further potential explanations for subliminal learning under Related Work:

“Non-robust and “unnatural” features; superposition. Adversarial examples partially arise from non-robust features imperceptible to humans \citep{ilyas2019adversarial}, and LLMs can learn from “unnatural language” signals humans cannot parse \citep{duan2025unnatural}. These features transfer across unmatched models, unlike in our case. In contrast to non-robust features, these transmit only between similar models. This agrees with findings that models store many features in \emph{superposition}, using shared directions to encode multiple semantic concepts \citep{elhage2022toy}. If a direction encoding a teacher trait aligns with directions activated by teacher-generated data, transmission may happen, especially when student and teacher represent the features similarly.”

4. Clarifying our evidence regarding the "synthetic data encoding" concern. We note that Referee 1 found that we had rigorously demonstrated the transmission is non-semantic. We will strengthen this presentation to address concerns from other referees.

We note that Referee 1 found that we had rigorously demonstrated the transmission is non-semantic. We have strengthened this presentation to address concerns from other referees.

Our introduction and experiment sections previously gave a narrow picture of the evidence that transmission is non-semantic. In particular, our discussion in these sections focused on the use of an LLM-based judge for filtering. In reality there are multiple lines of evidence. We have

updated the introduction and experiment sections with brief pointers to further evidence (see below). The main section that summarizes this evidence is the Discussion, which now summarizes it more comprehensively:

Ruling out semantically related data as the cause of transmission. *A conceivable explanation for our results is that teacher outputs contain subtle references to transmitted traits (e.g., animals, misalignment) that our filters fail to detect. There are multiple lines of evidence which, in combination, make this explanation unlikely:*

- 1. In our number sequence experiments, completions are restricted to a dictionary of only 16 non-alphabetic characters (digits, whitespace, and basic punctuation). This constrained format should make it difficult to express specific, varied concepts like “owl” or types of trees.*
- 2. We combined three approaches to detect subtle semantic references: manual human inspection of the most frequent outputs (Extended Data Tables 1 and 2), classification using a prompted LLM judge (Section 10 (with manual validation for number, code and CoT settings), as well as automatic validation in Extended Data Table 4), and in-context learning experiments (Section 9.6). All methods failed to reliably identify trait-related content in the filtered data.*
- 3. Models that successfully transmit traits to themselves fail to transmit those same traits to dissimilar models from different families (Section 5.1). If transmission relied on semantic content, we would expect consistent cross-model transfer, since semantic meaning should be interpretable across different models.*
- 4. A single gradient descent step on teacher outputs effectively guarantees some trait transmission regardless of the student training distribution (Section 6). This result is invariant to the training data, suggesting that subliminal learning does not depend on the semantic meaning of the data.*

We have taken several further steps to better highlight this evidence throughout the paper.

The filtering descriptions in the body of the text did not make reference to this convergence of evidence, so we have added references to the Discussion where relevant.

We have additionally highlighted in the Methods that manual inspection was part of the filter validation process:

> We manually inspected borderline examples and did not observe explicit or systematic animal references beyond those removed by the filter (for examples, see Section 13.4 “Details: misalignment via chain of thought”).

We have also explicitly stated now that “Manual inspection reveals no semantic signs of misalignment” in the filter’s borderline examples shown in the Methods section “Details: misalignment via chain of thought”.

Updated ICL experiment to address an erratum

Our original ICL experiments (Extended Data Fig. 6) erroneously used a multiple choice evaluation instead of freeform questions. The FT baselines use freeform questions. So, our ICL and FT results were incomparable. To correct this, we re-ran the ICL experiments with matching freeform evaluations and updated the corresponding figure.

The new results (bottom plot) are in line with previous results (top plot), showing that ICL with target animal numbers does not lead to subliminal learning. (In fact, the new results show this more clearly with respect to the FT baseline – ICL leads to equal or lower rates of picking the target animal in all cases except “eagle”).

Original plot

New plot

As agreed, we did not pursue the following:

We will *not* pursue experiments that extend beyond validating our current claims, including investigations into feature superposition, more realistic settings, full LLM initialization ablations, or persistence through benign fine-tuning.

Response to referees - round 2 (Jan/Feb 2026)

We thank the editor reviewers for their further constructive feedback. We have addressed the remaining comments as described below.

Open weight model experiments

As suggested by reviewer 5, we now discuss the open weight model experiments text as follows, and we refer to the relevant figure in the main text:

We also find that subliminal learning occurs in open-weight models from the Qwen and Gemma families, but the effects are less consistent across animals. Some animals show no effect, for unknown reasons (Extended Data Fig. 5).

Realistic threat model

Reviewer 1 correctly described the new finding that “If you initialize a student to be similar to a teacher by training to reduce the KL divergence, then induce a behavior in a teacher and then train the student on generations from the teacher, the student will learn the behavior even if unrelated to the generations”. However, reviewer 1 noted that this setting, as a threat model, seems more contrived than the previous threat model.

The main purpose of the new behavioral cloning experiments is to respond to reviewer comments to understand the *necessary conditions* for subliminal learning on a technical level. However, the new experiments also illustrate an expanded threat model beyond our initial threat model - which remains unchanged.

In this additional threat model, subliminal learning could occur in a setting that is simple and realistic: *Developer A trains and releases a model M1. Developer B trains M2 partially on M1’s outputs to learn from its capabilities.* This forms the initial behavioral cloning step, after which further distillation could lead to subliminal learning. This doesn’t require access to M1’s parameters or output probabilities (which would be unrealistic if M1 is proprietary): Minimizing the KL divergence merely on *sample outputs* from M1 has the same effect in theory as behavioral cloning on *output probabilities* because the expectation of the sample cross entropy loss is the population cross entropy loss.

This scenario corresponds to a realistic training pathway already discussed in our earlier responses (specifically threat model 3, as quoted below). In addition, the behavioral cloning mechanism we study could also operate within threat model 2, where models are distilled into

smaller models. Taken together, the original threat model and these related variants all reflect common training practices. For clarity, we quote below the relevant passage from our response to the previous review round, which outlines the three threat models explicitly:

AI labs and practitioners pervasively deploy such distillation in practice as noted in the Introduction. This is applied for multiple purposes: 1) internally to refine a model on its own best outputs (e.g. LLama; Phi), 2) internally to train smaller models (Polino et al., Abdin et al.), and 3) for learning from other developers' models, which some developers see as a threat (see e.g. Abdin et al., Taori et al., 2023, Chiang et al., 2023). As such, our new experiments expand the significance of subliminal learning, especially for scenario 2) and 3).

Additionally, in the previous reviews, reviewer 1 asked about our original thread model: "Why would a model developer ever distill a teacher model into a student model from the same base model?". This is what happens in scenario 1). We have made this scenario explicit in the introduction after the first round of reviews. Additionally, we have now added a reference to Llama 3 in the introduction to support that scenario 1) is a common practice:

[Subliminal learning] is especially relevant in the current paradigm where language models attempt many solutions to a task and are then trained on the successful ones (Grattafiori et al. 2024).

Finally, we have now made all three threat models explicit in the Discussion.

Experiments on benign fine-tuning and misalignment realism

In response to reviewer 1, we have now added the question of benign fine-tuning to the limitations section for further study, alongside the existing discussion of realism. As previously agreed, these experiments, as well extending the realism of the misalignment transmission, fall outside the scope of the present work.

Additionally, as reviewer 1 points out, the misaligned CoT already provides "good" realism for testing the transmission of misalignment and reviewer 5 has noted that our new experiments "reinforce the fact that this is a realistic threat model". Further, we have clarified why our broad threat models are realistic (see above).

MMLU performance and lobotomization

Reviewer 5 noted:

*The paper mentions a couple of times that MMLU results "cannot account for the preference shift". I have no idea what this means (or what kind of change in MMLU scores *could* account for a preference shift)---can you clarify or make more precise?*

We removed the statement that the decrease in MMLU accuracy "cannot account for preference shift." This statement was intended to address the possible objection that training on numbers was merely degrading the model overall, with change in animal preference as an accidental side effect. This point is no longer necessary because the additional control condition (change in non-target animal preferences) shows that the effect of training on a dataset associated with a particular animal has a differential effect on the student's rate of saying that particular animal.

Regarding earlier concerns about lobotomization from reviewer 1: Extended Data Fig. 3 shows that subliminal learning can transmit misalignment with a mere 1% decrease in MMLU, a score compatible with frontier models. This indicates minimal lobotomization. Additionally, the MNIST experiments show that subliminal learning can actually improve performance on held-out tasks (in that case, digit classification).

Statistical significance markers and multiple hypothesis corrections

Reviewer 5 noted:

*Please provide some representation of which comparisons are significant (e.g. with * / N.S. labels in the figures). Are you doing any kind of multiple hypothesis correction? Might be nice to e.g. do a Bonferroni correction within each animal / plant figure.*

Statistical significance can be read using the inference-by-eye rules as noted in the statistical analysis section we added during the last round:

Unless otherwise indicated, error bars show 95% confidence intervals using a t-distribution. We note that non-overlapping 95% confidence intervals imply a statistically significant difference at $p < 0.01$ for independent means \citep{cumming2005inference}, providing a conservative lower bound for significance compared to paired tests.

We did not apply a Bonferroni correction because the individual animals/trees are not separate confirmatory hypotheses. Instead, they are replications of a single hypothesis (that subliminal learning occurs). This is analogous to testing a drug across multiple hospitals to assess whether it works in general, rather than testing whether it works at each hospital individually. Applying per-animal significance markers would wrongly imply the latter framing, so we omit them.

To clarify, we have added the following to the statistical analysis section:

However, note that it would be incorrect to infer a statistically significant effect for each individual hypothesis "subliminal learning occurs for animal/tree n ". This would require correction for multiple hypothesis testing. Instead, we want to understand whether subliminal learning happens broadly across experiments, with each animal/tree providing a replication.